# Is Mitochondria DNA Variation a Biomarker for AD?

**DOI:** 10.3390/genes13101789

**Published:** 2022-10-03

**Authors:** Ruonan Gao, Suk Ling Ma

**Affiliations:** Department of Psychiatry, Faculty of Medicine, The Chinese University of Hong Kong, Hong Kong, China

**Keywords:** Alzheimer’s disease, mitochondria DNA, biomarker

## Abstract

Alzheimer’s Disease (AD) is the most prevalent form of dementia and is characterized by progressive memory loss and cognitive decline. The underlying mechanism of AD has not been fully understood. At present there is no method to detect AD at its early stage. Recent studies indicate that mitochondria dysfunction is related to AD pathogenesis. Altered mitochondria functions are found in AD and influence both amyloid-β (Aβ) and tau pathology. Variations in mitochondria DNA (mtDNA) lead to a change in energy metabolism in the brain and contribute to AD. MtDNA can reflect the status of mitochondria and therefore play an essential role in AD. In this review, we summarize the changes in mtDNA and mtDNA mutations in AD patients and discuss the possibility of mtDNA being a biomarker for the early diagnosis of AD.

## 1. Introduction

Alzheimer’s disease (AD) is a progressive neurodegenerative disease and it is the most common form of dementia among the elderly. In 2020, about 44 million people worldwide suffered from AD and this is estimated to double by 2050 [1]. Aging is the strongest risk factor for AD [2]. Prevalence doubles every five years among people over 60, rising from 1% in those between 60–64 to 40% in those over 85 years old [3]. AD is associated with a combination of genetic, environmental and lifestyle risk factors [4]. The major symptoms of AD include behavioral changes, personality changes, decline in memory, motor restlessness and destruction in speech and movement [5]. AD can be classified in three different stages: (1) A preclinical stage, which is a long asymptomatic period characterized by early pathological changes in the cortex and hippocampus [6]. Subjects at this stage will have a higher risk progressing to AD but have no obvious impact on their daily life or clinical manifestations [6,7]. (2) Mild cognitive impairment (MCI), which is characterized by several symptoms such as memory loss, disorientation, poor judgement, depression and being more impulsive. It is considered the earliest stage of the symptomatic cognitive decline before AD dementia [6]. Individuals perform worse in some cognitive domains and may take more time and make more mistakes compared to the past [8]. (3) The AD stage, in which the disease spreads to the cerebral cortex with the accumulation of neuritic plaques and neurofibrillary tangles. Patients have trouble in thinking and making decisions, difficulty in performing familiar tasks and increased memory loss and eventually progress to cognitive impairment [6].

At present, there is no treatment for full recovery from AD and only symptomatic treatment is available. Three major neuropathological hallmarks have been identified including extracellular amyloid plaques consisting of Aβ peptide, intracellular neurofibrillary tangles that consist of hyperphosphorylated tau, and neuronal loss. Currently there is no particular method to diagnose AD at an early stage [9]. Traditional AD diagnosis consists of clinical symptoms, neuroimaging such as computed tomography (CT), magnetic resonance imaging (MRI) and positron emission tomography (PET) of certain hallmarks, and cerebrospinal fluid (CSF) biomarkers, as well as AD-specific autoantibodies in the peripheral blood [10]. The AD pathological process may occur several years or even decades before clinical symptoms can be observed, and therefore, new diagnostic tools for the early detection of AD such as biomarkers are urgently needed.

## 2. MtDNA

Mitochondria are responsible for producing energy, detoxifying, communicating between brain cells, regulating temperature and keeping the redox balance of the cells [11,12]. In neurons, mitochondria participate in nerve signaling by maintaining neuron membrane potential. Most ATP consumed by the central nervous system (CNS) is supplied by oxidative metabolism, causing neurons fully rely on the normal function of mitochondria [11]. Recent studies have shown that mitochondrial dysfunction and increased oxidative stress are common features of AD and probably participate in the pathogenesis of AD, although the underlying mechanism is controversial [13]. Some studies have indicated that upstream pathologies, including Aβ, cause mitochondrial dysfunction, while others have suggested that mitochondrial dysfunction occurs independently of Aβ and directly disrupts brain functions or subsequently introduces other pathologies that lead to neurodegeneration [14].

The mitochondrial cascade hypothesis suggests that age-related mitochondria dysfunction is the primary cause of sporadic, late-onset AD, instead of tau and Aβ. Mitochondrial function influences amyloid precursor protein (APP) production, APP cleavage and Aβ accumulation. The dysfunction of mitochondria may lead to other AD-associated molecular alterations, such as increased oxidative stress, affected tau phosphorylation and inflammation [15]. Mitochondria deficiency in AD brains explains the increased Aβ production [16]. In the early stage of sporadic AD, mitochondrial dysfunction can be found before the formation of Aβ or tau, which is related to proteostasis, and, in turn, promotes the aggregation of Aβ and tau [11]. Another hypothesis assumes AD is the result of an attempt to preserve neuronal integrity via metabolic reprogramming under mitochondrial dysfunction. The failure of this compensatory mechanism results in neuronal death and subsequent neurodegeneration [17].

Several mitochondrial alterations are found in AD (Figure 1). Mitochondrial bioenergetic function reduces, including decreased respiratory chain activity, ATP production and enzymes involved in the mitochondrial tricarboxylic acid cycle, while the content of free radicals and reactive oxygen species (ROS) are elevated. Changes in mitochondrial dynamics and distribution, including the dysfunction of mitochondrial axonal transport, dysregulated organellar dynamic and a closer localization to the nucleus are also observed in AD cells. In the hippocampal CA1 region of an AD patient, mitochondria are distributed around the nucleus of neurons [18]. Impaired metabolic activity and oxidative phosphorylation are found in a 3xTg mouse model of AD, as evidenced by the decreased activity of mitochondrial respiratory chain complex IV and the expression of complex I–V. These impairments may contribute to reduced mitochondrial bioenergetics and ultimate degeneration [19].

There is a connection between mitochondria and Aβ pathology (Figure 1). Incubating cells and mitochondria with Aβ has a negative impact on mitochondrial functions, including respiration, protein import, organellar transport, organellar localization and organellar dynamics [18]. APP, Aβ and components of the γ-secretase complex localize closely to mitochondria and can be found in mitochondrially-enriched fractions of the brain samples, indicating mitochondria may directly participate in AD [20].

Studies have also shown an association between tau pathology and mitochondrial dysfunction (Figure 1). Hyperphosphorylated and cleaved tau increase the amount of stationary mitochondria by increasing the intermicrotubular distance. Neurons expressing tau show decreased anterograde transport and increased retrograde transport, leading to a perinuclear distribution of mitochondria. The overexpression of phosphorylated tau increases mitochondrial length and decreases fission proteins. Tau pathology also influences mitochondrial membrane potential and increases oxidative stress, eventually raising the sensitivity to different molecules, such as Aβ [21].

Human mtDNA is a circular double-stranded DNA containing 16,569 base pairs. MtDNA consists of 37 genes, including 2 ribosomal RNA (rRNA) genes, 22 transfer RNA (tRNA) genes for mtDNA translation and 13 genes encoding proteins of the mitochondrial respiratory chain [22,23]. MtDNA also owns a non-coding region called displacement loop (D-loop) region, which mainly participates in the process of mtDNA replication and transcription [24]. Single nucleotide polymorphisms (SNP), deletions, insertions and copy number variations of mtDNA can lead to alteration of energy metabolism in brain and result in neurodegenerative diseases [13,25].

MtDNA is hypothesized to be involved in the pathogenesis of AD. In late-onset Alzheimer’s disease (LOAD) patients’ brain, the oxidative phosphorylation complex component, complex activity and energy production are significantly decreased. Mitochondrial dysfunction leads to a higher level of ROS and damage to mtDNA. APP cleavage is changed due to mitochondrial dysfunction and this results in the accumulation of Aβ [26]. Under cellular stress and damage, mtDNA can either be captured into vesicles or exosomes to maintain normal cellular function, or it will be released at a low concentration as circulating cell-free mtDNA (ccf-mtDNA). However, under a high concentration of ROS, the oxidized mtDNA fragment can be released through large pores in the outer membrane of mitochondria. Released mtDNA may be regarded as a damage-associated molecular pattern (DAMP) or an antigen and induce subsequent inflammatory activation through TNFα, the TLR9/NF-kB signaling pathway or the NLRP3 inflammasome [27,28]. In cybrid cell lines transferred with mtDNA from AD patients, complex IV activity was significantly reduced and ROS production was increased, suggesting the role of mtDNA in the pathology of AD [29]. Changes in mtDNA may reflect the status of mitochondria. Since mtDNA is more resistant to nuclease degradation, and it can be stably detected in both plasma and CSF [30], this makes mtDNA a potential biomarker for AD.

## 3. MtDNA Mutation

MtDNA has a higher mutation rate when compared to nuclear DNA because it does not have protective histones and it is closer to the inner membrane where ROS are generated [31,32]. The mutation rate of mtDNA is 10- to 200-fold higher than nuclear DNA under oxidative stress [30]. Moreover, mtDNA replicates more frequently than nuclear DNA and the repair efficiency for mtDNA is lower than nuclear DNA [33]. The accumulation of mtDNA mutations during lifetime is an important contributor to neurogenerative diseases [34]. MtDNA mutations can affect coding properties and normal cellular metabolism. They may cause damage to the electron transport chain, leading to neurodegeneration [35]. Aβ can promote oxidative stress, which subsequently leads to mtDNA mutations and mitochondria dysfunction [36]. Several mutations and deletions in mtDNA have been associated with AD (Figure 2).

In the frontal cortex of AD patients, the level of the common 4977np mtDNA deletion was 15-fold higher when compared with controls but declined with age to one-fifth after 75 years old [37]. The decrease in the amount of mtDNA deletions in AD patients with older age might be due to apoptosis of the brain cells containing the most mutated mtDNA [38]. It is suggested that fragment deletion is a result of faulty replication between direct repeats of mtDNA [26]. The exact mechanism for the fragment deletion in mtDNA has not been fully understood and may be different depending on the age of the patient or the position of the deletion.

Cytochrome c oxidase (CO) is kinetically abnormal in LOAD. Five missense mutations in CO catalytic centers, including CO1 G-A (Val-Ile) at position 6366, CO1 A-G (Thr-Ala) at position 7146, CO2 C-T (Thr-Ile) at position 7650, CO2 C-T (Leu-Phe) at position 7868 and CO2 A-G (Ile-Val) at position 8021, were increased in AD patients compared with controls [39]. These five mtDNA mutations lead to amino acid changes and may decrease CO catalytic activity by distorting the secondary structure of the relevant area [40] and increase the production of reactive oxygen species, and therefore participate in subsequent events that result in AD [39]. The other two missense mutations, G-T at np 8206 (Met-Ile) and A-T at np 8224 (Leu-Phe), were also found in the CO gene sequence from AD patients [41]. A tRNA Gln gene variant A-G at nucleotide pair 4336 was found to be higher in Caucasian AD patients than controls [42,43], and the result was replicated in another small cohort from Germany [44]. It was hypothesized that the 4336G mutation caused impairments in complex I and damage to neurons [43]. However, the findings were not replicated by other groups [45,46]. In a study that sequenced 22 mitochondrial tRNA, two mutations at positions 709 (rRNA 12S) and 15,928 (tRNA (Thr)) showed a significantly higher frequency in controls compared to AD patients, suggesting a protective role in AD [47]. NADH dehydrogenase subunit 2 (ND2) point mutation G5460(A/T) at codon 331 was found to be significantly higher in AD brains [48]. However, the result was not replicated by other group [45]. Two studies using post-mortem samples, blood samples and human brain tissues from AD patients also failed to confirm the high prevalence of the 5460(A/T) mutation in AD [49,50]. Alterations in mtDNA modifications 3197 T-C of mtDNA NADH dehydrogenase subunit 1 (ND1) gene, 3338 T-C of 16S rRNA mtDNA gene and 3199 T-C of the 16S rRNA mtDNA gene were found in Portuguese AD patients [51]. However, the result in 16S rRNA (nt. 3196) and ND1 (nt. 3397) was not found in Japanese sporadic AD [46]. There might be ethnic differences on the effect of mtDNA mutations, and further studies will be required to confirm the application of mtDNA mutations as markers for the early detection of AD.

As well as mRNA, rRNA and tRNA genes, mtDNA also contains a 1112-nucleotide pair control region (CR). T414G mutation in CR was found to be 73% higher in AD brains but was absent from the controls; T414C and T477C mutations were found in certain AD patients’ brains, and several AD patients’ brains had additional CR mutations of T146C and T195C [52]. These CR mutations might explain the mitochondrial dysfunction in AD. In AD brains of patients aged 55 to 90 years, the frequency of the mtDNA regulatory control region (RCR) mutations was more than two-fold higher than in control brains. The frequency of RCR mutations in serum samples and lymphoblastoid cell lines of 81- to 95-year-old AD patients was 2.5 times higher than age-matched controls. All these data suggest the increased frequency of mtDNA mutations is a systemic phenomenon in patients with AD [53].

No clear evidence has shown that mtDNA mutations are the primary cause of AD at present. It is more likely that mtDNA mutations occur as a secondary result of the primary disease process. When there is a high level of cells containing mtDNA mutations, they may lead to AD pathologies and result in disease progression in AD [31].

Different mtDNA haplogroups can be defined by a combination of different mtDNA SNPs in the mitochondrial genome [54]. Some haplogroups have been associated with AD. African American carriers of haplogroup L1 had a higher risk of dementia and lower Aβ42 levels compared with the most common African haplogroup L3. The m.5046G-A mutation encoding a p.V193I, ND2 substitution was associated with increased Aβ42 among haplogroup L3 [55]. Haplogroup H was significantly more common among the Asturias (northern Spain) LOAD patients [54]. Consistent with the above result, sub-haplogroup H5 and sub-haplogroup HV showed a higher risk of AD in Italians and Polish people, respectively [56,57]. Caucasian Alzheimer’s Disease Neuroimaging Initiative (ADNI) subjects showed that the haplogroup UK had a higher genetic susceptibility to AD [58]. MtDNA haplogroup J was more commonly found in AD patients than in controls [59]. A study found that haplogroup J carriers had lower baseline performance and improved slowly in verbal memory tests when compared to haplogroup H carriers [60]. In Mexican Americans, the mtDNA 8oxoG mutation load was significantly higher when compared to non-Hispanic whites. Haplogroups A and C exhibited higher 8oxoG variant counts, while haplogroups I and K exhibited lower 8oxoG variant counts [61]. All these findings suggest that mtDNA haplogroups are associated with the risk of AD, but it is highly specific to populations. More research is required to elucidate the ethnic differences in the importance of mtDNA haplogroups to the risk of AD.

A study also showed that both AD patients and healthy elderly had a higher burden of brain mtDNA point mutations than younger controls, as detected by polymerase chain reaction (PCR) cloning sequencing. However, no significant difference in mtDNA point mutations between AD patients and elderly controls or among different cortical regions was found. Most of the mutations are rare [62]. In a study measuring the mtDNA mutation rate and spectrum in AD patients, and cognitive controls and high-pathology controls (hpC) using a highly sensitive random mutation capture (RMC) assay, the result indicated that mtDNA mutation rates were similar among the three groups in both the temporal cortex (TC) and cerebellum (CE) [63].

Both sub-haplogroups defined by ancient polymorphisms and the accumulation of sporadic mutations are probably involved in AD [56]. The effect of mtDNA mutation and mtDNA haplotype towards AD are controversial among different studies. At present, there is no clear conclusion as to which genotype has an impact on AD or the underlying mechanism showing how mtDNA genetic variation correlates with AD pathogenesis. MtDNA mutation reflects the cellular changes under disease conditions, especially oxidative stress. Therefore, mtDNA mutation can be used as a biomarker for AD detection or as a risk factor for AD prediction.

## 4. MtDNA Copy Number

Human mtDNA is present in multiple copies per cell. MtDNA copy number (mtDNAcn) analysis is the most commonly used method to determine mitochondria abundance [9]. The alteration in mtDNAcn is considered as an indicator for damaged mitochondria [64]. The mtDNAcn correlates with cellular energy production and metabolism, and, therefore, may differ greatly between patients and controls [65].

MtDNA/nuclear DNA ratio of the ND2 and 18S rRNA gene copy numbers showed a 50% reduction in cellular mtDNAcn in the frontal cortex of AD patients [52]. Brain tissue from neuropathologically characterized post-mortem samples showed a significantly lower mtDNAcn in AD than controls [66]. Hippocampal pyramidal neurons from AD patients had lower mtDNAcn in comparison with neurons from controls. However, mtDNAcn in all the three mentioned cell types were not associated with hippocampal Aβ_1-42_ levels [67]. MtDNA from the blood of AD patients, and the hippocampus and CE of AD necropsies, did not show differences compared with controls. AD patients had a 28% lower amount of mtDNA in the frontal cortex. As the frontal cortex was severely impaired in AD, the results indicated that mitochondrial loss may play a role in the pathogenesis of AD [68]. Whole genome sequencing (WGS) showed a lower level of mtDNAcn in the dorsolateral prefrontal cortex (DLPFC) and posterior cingulate cortex (PCC) in AD. Differences in mtDNAcn in the CE were not significant between AD and controls. Data from the TC in Mayo and frontal pole (FP) in MSBB also confirmed the result. Three studies showed a reduction in mtDNAcn by 7.0–14.2% in AD when compared with controls in different cortical regions [69]. Significant mtDNA depletion was also found in the TC from AD patients when compared to controls and high-pathology controls (hpC), but no variation in mtDNAcn among these three groups was observed in CE [63]. When considering the association between mtDNAcn and 10 common pathologies of AD, in the DLPFC, mtDNAcn was mainly associated with tau pathology and cognitive decline. In the PCC, an association between TDP-43 pathology and cognitive decline was found. These data showed that mtDNAcn was negatively correlated with AD pathology and positively correlated with cognitive performance. When regressing cognitive function was an outcome, mtDNAcn showed a significant association with cognitive function. Furthermore, a missense mtDNA SNP rs11085147 of the gene Lon Peptidase 1 (LONP1) was significantly associated with mtDNAcn in four different regions in the brain [69].

CSF and the extracellular space of the brain are in direct contact, and it can reflect changes in the brain intuitively. Therefore, CSF is the most representative source of AD biomarkers [70]. The level of cell-free mtDNA (cf-mtDNA) in CSF from asymptomatic patients at risk of developing AD and symptomatic AD patients was significantly lower compared with the controls. Young, presymptomatic subjects carrying the pathogenic PSEN1 mutation had a lower level of CSF cf-mtDNA as measured by quantitative real-time PCR (qPCR). The CSF mtDNA level distinguished AD patients from controls with high sensitivity and specificity as shown by the receiver operating characteristic (ROC) curve. The same result was validated in another cohort by using droplet digital PCR (ddPCR). These results indicate that a decrease in the mtDNA level in AD appears earlier than the common biomarkers t-tau and p-tau can be detected [71]. Low CSF mtDNA is a potential biomarker for the detection of AD at an early stage.

A study showed that the disease progression of AD was associated with the level of cf-mtDNA in CSF. A significant decrease in cf-mtDNA in CSF for AD patients who progressed faster when compared to controls was reported as measured by mtDNA copies per microliter CSF. The CSF level of cf-mtDNA showed a positive correlation with Aβ and a negative correlation with p-tau, but it did not have any correlation with t-tau in AD patients. The cf-mtDNA/p-tau ratio was associated with a diagnosis of AD, with sensitivity and specificity of over 90%. These results indicate that the cf-mtDNA level might be a good biomarker for early AD diagnosis and low CSF mtDNA together with typical biomarkers, low Aβ and high p-tau, may increase the sensitivity and specificity of the test [72]. However, a later study using ddPCR to quantify the absolute CSF mtDNA of AD patients failed to replicate the result. The study showed an increase in mtDNA levels in AD patients compared with controls. MtDNA levels ranged widely among individuals of the same diagnostic group, leading to a high degree of overlap between AD patients and controls. The ROC curve showed a moderate ability of mtDNA to distinguish the two groups [73]. Therefore, further studies will be required to optimize the cutoff value for clinical application.

In situ analysis of mtDNA showed the significantly higher mtDNA level was mainly restricted to large vulnerable neurons of the hippocampus and neocortex in AD brain tissues, as with pyramidal neurons. The change in mtDNA in specific vulnerable neurons was small when considering the whole region, and this may explain why some studies using tissues representing a variety of cell types failed to detect an increase in mtDNA level [74].

MtDNAcn alterations can also be found in blood cells. Peripheral blood mononuclear cells’ (PBMCs’) mtDNA content was significantly reduced in AD and MCI patients when compared with controls. The loss of mtDNA content was also correlated with a lower Mini-Mental State Examination (MMSE) score, indicating cognitive decline. The ratio between mtDNAcn and nuclear DNA copy number increased as a result of mitochondria biogenesis under oxidative stress. However, persistent high oxidative stress may cause mtDNA depletion and mitochondrial dysfunction due to mtDNA and protein damage [75]. Another study focusing on PBMC showed lower levels of mtDNAcn in AD patients and the increased frequency of mtDNA deletions in peripheral leukocytes compared with the controls. Leukocytotic mtDNAcn was negatively correlated with the risk of development of AD; a higher level of leukocytotic mtDNAcn reduces the possibility of AD. These results indicated that AD patients had weaker mtDNA function [76]. MtDNAcn in the peripheral leukocytes was also negatively correlated with the number of the APOE4 allele in an AD cohort [77].

OXPHOS genes for subunit complexes I, III, IV and V encoded by mtDNA were significantly increased in MCI and AD blood cells compared with controls in mitochondrial transcripts (MT) MT-ND1, MT-CO1, MT-CO2, MT-CO3 and MT-ATP6. There was also a significant increase in MT-ND2, MT-ND5 and MT-ND6 gene expression in MCI subjects. No significant difference in mtDNAcn between the control, MCI and AD groups was found, suggesting no change in the steady-state numbers of mitochondria [78]. This result was not particularly informative due to the heterogeneity among different blood cell types. A further study isolated different blood cell types according to their surface markers [9]. In separated CD4+, CD19+ and CD56+ peripheral lymphocytes, the mtDNA fold change in the early and late-stage AD patients was lower compared with controls. The mtDNA level in CD8+ peripheral lymphocytes was significantly reduced in the late-stage AD patients only. The mtDNA levels in CD4+ and CD19+ peripheral lymphocytes were almost the same in both early- and late-stage AD patients. The mtDNA level in CD56+ peripheral lymphocytes in the late-stage AD patients was higher than AD patients in the early stage [9].

In an amyloid precursor protein/presenilin 1 (APP/PS1) transgenic mice model, the mtDNAcn and mitochondrial gene expression of 12S rRNA were reduced compared with wild type C57BL/6J mice [79].

MtDNAcn variation can also be detected in the early stage of AD. Lower mtDNAcn was associated with worse cognitive performance in non-Hispanic white individuals and associated with cognitive impairment (either MCI or AD) in Mexican Americans [80,81]. MtDNA in frontal, parietal and cerebellar regions of post-mortem human brains from normal controls, MCI and AD cases showed significant regional variation, with the highest in the parietal cortex and lowest in the CE. In MCI and AD patients without diabetes, the parietal cortices had decreased mtDNA compared with controls [82]. In apparently healthy elderly women, mtDNAcn was positively correlated with the MMSE score. Cognitive impaired individuals (MMSE score < 25) had lower mtDNAcn compared to controls [83]. MtDNAcn in combination with other biomarkers can be used to predict AD. In a study conducted in Korea, mtDNAcn and telomere length were used to monitor cognitive decline in the elderly. MtDNAcn decreased significantly in the cognitive dysfunction group as compared with controls. People with high telomere length/high mtDNAcn were more likely to have cognitive dysfunction than people with low telomere/low mtDNAcn [84].

Many studies have investigated the changes in mtDNAcn in AD but have controversial conclusions (Table 1). No consensus has been reached so far. In individuals with cognitive impairment, mtDNA was found to be decreased in a few studies, indicating its potential role to detect AD at an early stage. In AD patients, most studies have shown that mtDNAcn is lower or has no significant changes compared with controls. The possible reason may be due to decreased mitochondria activity or impaired mitochondria function in AD patients, which therefore reduces the rate of mtDNA replication. MtDNAcn decreases in the frontal cortex, DLPFC, PCC and TC in AD brains but shows no change in the CE. There were no conclusive findings on mtDNAcn in the hippocampus, so further studies will be required to confirm the result. Based on the observed changes in mtDNAcn in AD patients, it might serve as an AD biomarker with further investigation.

## 5. Discussion

With the aging population and the exponential increase in the prevalence of AD, there is an urgent need for the identification of biomarkers for early diagnosis. Evidence showed that mitochondrial dysfunction is associated with AD. MtDNA directly reflects mitochondria variations and therefore it may be developed as a biomarker. Some studies showed that mtDNA level was increased in AD, which may be due to neuron death and apoptosis, resulting in the release of mtDNA to extracellular space. On the other hand, some studies reported a reduced level of mtDNA in AD patients, which may be explained by the decreasing metabolic activity and lower energy required in such a condition. Controversial findings may also be due to different mtDNA extraction methods, experiment methods, AD diagnostic criteria, sample sources and study designs among different studies.

MtDNA content can be quantified by many different methods. qPCR is the most widely used method to quantify relative or absolute mtDNA content. Many studies assess the relative mtDNAcn as a ratio between mtDNA level and level of a selected nuclear gene, making comparison between studies very difficult. Recent advances in next generation sequencing (NGS) enable the high-throughput, high-sensitivity and accurate measurement of the mtDNA mutation load and mtDNAcn in large-scale data sets, which provide further opportunity to discover the effect of these factors on AD [85].

MtDNA varies differently among cell types and brain regions. The cellular composition of the tissue under investigation should be taken into consideration [85]. Samples from the brain and blood may show different results; mtDNA from different brain regions, whole blood samples or specific isolated cell types would also be different. As different regions of the brain have specific metabolic requirements and are affected to different extents in AD, mtDNA changes may be different [82]. Biomarkers in blood will be promising as it is non-invasive and simple.

MtDNA changes are associated with disease progression. The controversial results from different studies might have arisen from the diagnosis and case classification. In addition, some studies focused on clinical symptoms and results from neuropsychological examination, while others focused on the histopathological hallmarks of AD. These factors might contribute to the inconsistent findings among different studies.

Other genetic and environmental factors should also be considered to raise the reliability of the results [85]. AD animal models will be useful for elucidating the cause and-effect relationships [85]. Longitudinal studies are needed to detect changes in mtDNA along disease progression. Larger cohorts including different races and mtDNA haplotypes are also needed to investigate the influence of genetic variations on mtDNA.

## 6. Conclusions

This review focuses on the potential ability of mtDNA to be a biomarker for AD detection. As mitochondria are involved in the pathogenesis of AD, mtDNA alteration can reflect some disease-related problems. Large-scale studies including different ethnic groups might be required to elucidate the association between the changes in mtDNA and the disease progression of AD. In addition, cell models and animal models will play an important role in understanding the relationship between mtDNA changes and pathological changes in AD. In conclusion, several mtDNA changes including the mtDNAcn, mtDNA content and mtDNA mutation have been identified as promising diagnostic biomarkers.

## Figures and Tables

**Figure 1 genes-13-01789-f001:**
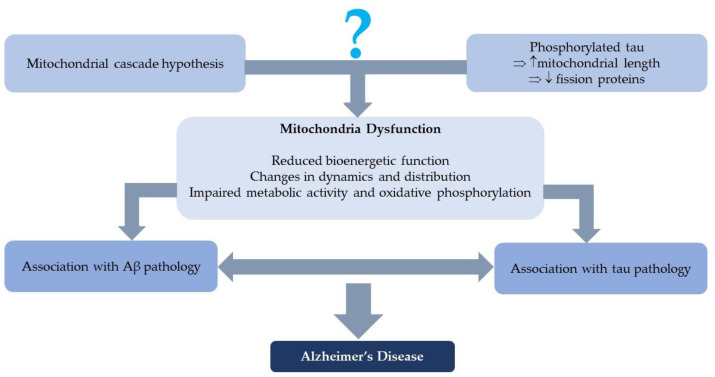
The role of mitochondria dysfunction in AD.

**Figure 2 genes-13-01789-f002:**
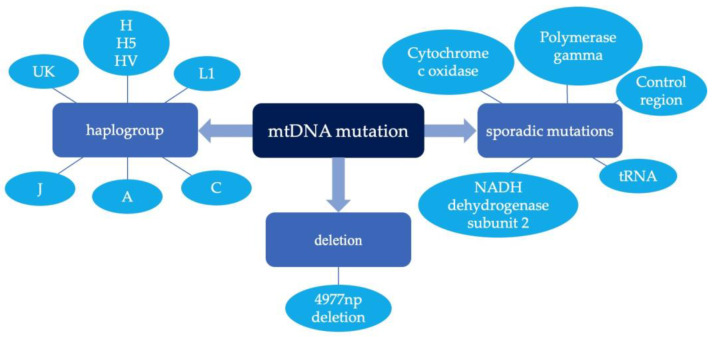
MtDNA mutations in Alzheimer’s disease (AD).

**Table 1 genes-13-01789-t001:** MtDNA changes in AD.

Source	Tissue/Cell Type	Technique	Ratio	Trend	Disease Stage	Reference
Brain			mean mtDNA read depth/mean exome read depth	decrease	AD	[66]
Brain	Hippocampus	Multiplex qPCR	comparing to a standard curve of circular human mtDNA run on the same plate	decrease	AD	[67]
Brain	Frontal cortex	RT-PCR	mtND2/r18S	decrease	AD	[68]
Brain	Hippocampus and CE	RT-PCR	mtND2/r18S	No change	AD	[68]
Blood		RT-PCR	mtND2/r18S	No change	AD	[68]
Brain	Frontal cortex	qRT-PCR	mtND2/r18S	decrease		[52]
Brain	DLPFC, PCC	WGS	median sequence coverages of the autosomal chromosomes covnuc and of the mitochondrial genome covmt (covmt/covnuc) × 2	decrease	AD	[69]
Brain	CE	WGS	median sequence coverages of the autosomal chromosomes covnuc and of the mitochondrial genome covmt (covmt/covnuc) × 2	No change	AD	[69]
Brain	TC	ddPCR	mitochondrial (Walker)/nuclear (RPP30) loci	decrease	AD	[63]
Brain	CE	ddPCR	mitochondrial (Walker)/nuclear (RPP30) loci	No change	AD	[63]
Brain	CSF	qPCR; ddPCR	1 copy of mtDNA corresponds to 18.16 attogram	decrease	symptomatic AD	[71]
Brain	CSF	ddPCR	copies/µL of CSF	decrease	AD patients progressed faster	[72]
Brain	CSF	ddPCR	copies/µL of CSF	increase	AD	[73]
Brain	Pyramidal neurons	In situ hybridization		increase	AD	[74]
Blood	PBMC	qRT-PCR	mtDNA/a reference single copy gene	decrease	MCI and AD	[76]
Blood	Leukocyte	qRT-PCR	mtDNA/a reference single copy gene	decrease	AD	[76]
Blood	CD4+, CD19+ and CD56+ peripheral lymphocytes	RT-PCR	mtDNA/β globin	decrease	early- and late-stage AD	[9]
Blood	CD8+ peripheral lymphocytes	RT-PCR	mtDNA/β globin	decrease	late-stage AD	[9]
Blood	CD56+ peripheral lymphocytes	RT-PCR	mtDNA/β globin	decrease	early- and late-stage AD	[9]
APP/PS1 transgenic mice model		qRT-PCR	12 S rRNA/18 S rRNA	decrease		[79]
Brain		WGS	mitochondrial genomes/nuclear genome	decrease	cognitive impaired	[80]
Blood	Buffy coat	RT-PCR	nuclear DNA/mtDNA	decrease	MCI and AD	[81]
Brain	Parietal cortex	qRT-PCR	mtDNA/β-2-microglobulin	decrease	MCI and AD	[82]
Blood		RT-PCR	mtND1/β globin	decrease	cognitive impaired	[83]
Blood		RT-PCR	mtDNA/β globin	decrease	cognitive dysfunction	[84]

Abbreviations: quantitative polymerase chain reaction (qPCR); real-time polymerase chain reaction (RT-PCR); quantitative real-time polymerase chain reaction (qRT-PCR); digital droplet polymerase chain reaction (ddPCR); whole genome sequencing (WGS); cerebrospinal fluid (CSF); cerebellum (CE); temporal cortex (TC); dorsolateral prefrontal cortex (DLPFC); posterior cingulate cortex (PCC); peripheral blood mononuclear cell (PBMC); mild cognitive impairment (MCI); Alzheimer’s disease (AD).

## Data Availability

Not applicable.

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
