# Peer review of "Is Mitochondria DNA Variation a Biomarker for AD?"

_genes, 2022, doi:10.3390/genes13101789_

Round 1

Reviewer 1 Report

Summary of review

This study focuses on mitochondrial DNA (mtDNA) and discusses the potential of mtDNA as a biomarker for the early diagnosis of Alzheimer's disease (AD). In the early phase of sporadic AD, mitochondrial dysfunction can be found prior to forming Aβ or tau, which has been associated with proteostasis and promotes the damaging neuron aggregation of proteins such as Aβ and tau. Changes in mitochondrial dynamics and distribution, including dysfunction of mitochondrial axonal transport, dysregulated organellar dynamic, and a closer localization to the nucleus is also observed in AD cells, indicating mitochondria may be involved in AD. Currently, there is no specific test to diagnose AD at an early stage. Changes in mtDNA reflect mitochondria's status; therefore, mtDNA is a potential biomarker for AD.

Comments and Suggestions for Authors

The following points need to be considered in future versions:

1.     The paper should use a graphic presentation when the authors explain the mitochondria’s role in AD.

2.     Authors need to rationalise why most studies showed that mtDNAcn of AD patients is lower or has no significant changes compared with controls. This biomarker's diagnostic and predictive role is not fully understood in non-AD individuals; therefore, mtDNAcn in blood samples and certain regions of the brain can be developed as an AD biomarker is a self conductorly statement.

3.     The current manuscript form has many grammatical mistakes like conjugation, punctuation, and the wrong tense. Also, some sentences are unclear, unnecessary wordy, and hard to follow and need to increase their readability. The manuscript needs extensive English editing by a native speaker, and it has 34% similarity  with the published content which was identified by ithentcate software.

4.     An extensive review of literature is required as many recent studies linking  mtDNA and AD are missing.

Author Response

1. The paper should use a graphic presentation when the authors explain the mitochondria’s role in AD.
Figure.1 is added in the revised manuscript to illustrate the role of mitochondria in AD.
2. Authors need to rationalise why most studies showed that mtDNAcn of AD patients is lower or has no significant changes compared with controls. This biomarker's diagnostic and predictive role is not fully understood in non-AD individuals; therefore, mtDNAcn in blood samples and certain regions of the brain can be developed as an AD biomarker is a self conductorly statement.
Further explanation on the decrease of mtDNAcn in AD was included in the revised manuscript.
3. The current manuscript form has many grammatical mistakes like conjugation, punctuation, and the wrong tense. Also, some sentences are unclear, unnecessary wordy, and hard to follow and need to increase their readability. The manuscript needs extensive English editing by a native speaker, and it has 34% similarity with the published content which was identified by ithentcate software.
The manuscript is revised accordingly
4. An extensive review of literature is required as many recent studies linking mtDNA and AD are missing.
Some more publications are added in the revision to illustrate the linkage between mtDNA and AD.

Reviewer 2 Report

The review presents a wealth of studies on mitochondria in AD. However, several problems require attention.

1) The review appears as a sum of sentences on a specific topic, often wholly independent from the previous and the following. This effect is particularly true in the Abstract and recurs throughout all sections. Authors should try their best to harmonize the sentences and the topics they are dealing with.

2) an appropriate description of AD in terms of clinical presentation and evolution should be given. The authors rely on a paper describing specific aspects of AD symptoms (ref 5). Still, they miss providing clear-cut information on the typical clinical presentation of the disease, a description of the early AD (MCI), the clinical evolution with spreading to the whole cortex, etc. AD is a complex disease and merits an appropriate description. I would suggest enlarging and improving the introduction accordingly

3) The section “MtDNA” describes mitochondria and their putative role in AD. Only in the last paragraph do authors start writing about mtDNA. A vignette figure explaining the concepts highlighted in this section is necessary.

4) In Sections 3 and 4, the authors proceed to a complicated and challenging presentation of controversial data on mtDNA mutations and mtDNA CV in AD. Reading is hard; the flow is interrupted, often monotonous. Authors should try their best to rewrite and improve these two sections, which should be enriched with illustrations/vignettes.

5) In summary, the review highlights only controversial data, often in the contradiction between them, and the authors underline this problem by opening the discussion.  Still, the authors must highlight those indisputable proofs and evidence of the role of mitochondria in AD, especially as biomarkers, as highlighted in the title.

6. A minor point: English style should be improved 

Author Response

1.The review appears as a sum of sentences on a specific topic, often wholly independent from the previous and the following. This effect is particularly true in the Abstract and recurs throughout all sections. Authors should try their best to harmonize the sentences and the topics they are dealing with.
The authors revised the manuscript accordingly.
2. an appropriate description of AD in terms of clinical presentation and evolution should be given. The authors rely on a paper describing specific aspects of AD symptoms (ref 5). Still, they miss providing clear-cut information on the typical clinical presentation of the disease, a description of the early AD (MCI), the clinical evolution with spreading to the whole cortex, etc. AD is a complex disease and merits an appropriate description. I would suggest enlarging and improving the introduction accordingly .
The authors appreciated the reviewer’s comment and related information is added in the revised manuscript.
3. The section “MtDNA” describes mitochondria and their putative role in AD. Only in the last paragraph do authors start writing about mtDNA. A vignette figure explaining the concepts highlighted in this section is necessary.
More information regarding mtDNA and a figure were added in the revised manuscript according to the reviewer’s comment.
4. In Sections 3 and 4, the authors proceed to a complicated and challenging presentation of controversial data on mtDNA mutations and mtDNA CV in AD. Reading is hard; the flow is interrupted, often monotonous. Authors should try their best to rewrite and improve these two sections, which should be enriched with illustrations/vignettes. A figure was added in the revised manuscript according to reviewer’s comment .
5. In summary, the review highlights only controversial data, often in the contradiction between them, and the authors underline this problem by opening the discussion. Still, the authors must highlight those indisputable proofs and evidence of the role of mitochondria in AD, especially as biomarkers, as highlighted in the title.
The content is revised accordingly in the revision.
6. A minor point: English style should be improved.
The content is revised accordingly in the revision.

Reviewer 3 Report

The paper focuses on the possibility of mtDNA to be a biomarker for AD. It’s a topic of interest to the researchers in the related areas but the paper needs very significant improvement before acceptance for publication. My detailed comments are as follows:

1.The title is Mitochondria dysfunction as biomarker for Alzheimer’s disease”, but this paper only summarized the mitochondrial DNA (mtDNA) problem in Alzheimer’s disease (AD).

2.Relevant research background of mitochondrial & mtDNA needs to be supplemented in INTRODUCTION. 

3. In page 3, MtDNA, the authors’ description of the relationship between mtDNA & AD is not detailed enough and too short. It’s necessary to figure the pathogenic mechanism between mtDNA & AD.

4. Specific mtDNA alterations, if any, and a potential causal role of mtDNA alterations in AD pathogenesis are yet to be proven. Additionally, mtDNA alterations are found in other neurodegenerative diseases but not specific to AD, how to prove mtDNA can to be a biomarker for early detection of AD?

6.The description of each section about mtDNA alterations is too long and lacks the author’s personal understanding.

7.CONCLUSIONS needs more in it, as it’s more of an afterthought. The authors are suggested to highlight important findings and include afterthought of this work.

Author Response

1.The title is “Mitochondria dysfunction as biomarker for Alzheimer’s disease”, but this paper only summarized the mitochondrial DNA (mtDNA)problem in Alzheimer’s disease (AD).
Title was revised.
2. Relevant research background of mitochondrial & mtDNA needs to be supplemented in INTRODUCTION. Mitochondria and mtDNA information was added in introduction part of the revision.
3. In page 3, MtDNA, the authors’ description of the relationship between mtDNA & AD is not detailed enough and too short. It’s necessary to figure the pathogenic mechanism between mtDNA & AD.
Possible mechanism between mtDNA and AD was added in the revised manuscript.
4. Specific mtDNA alterations, if any, and a potential causal role of mtDNA alterations in AD pathogenesis are yet to be proven. Additionally, mtDNA alterations are found in other neurodegenerative diseases but not specific to AD, how to prove mtDNA can to be a biomarker for early detection of AD?
This review is trying to list out all the possible evidence that mtDNA variation might be a possible marker for AD and the finding is not conclusive yet, therefore more research is required. This view is added in the revised manuscript.
5. The description of each section about mtDNA alterations is too long and lacks the author’s personal understanding. The manuscript is revised accordingly.
6. CONCLUSIONS needs more in it, as it’s more of an afterthought. The authors are suggested to highlight important findings and include afterthought of this

Round 2

Reviewer 1 Report

I am satisfied with the revised mansucript and I reccommend the same for the publication

Author Response

Thank you!

Reviewer 2 Report

Authors included appropriate figures and impoved significantly the manuscript
